# Effectiveness of Repetitive Transcranial Magnetic Stimulation Combined with Visual Feedback Training in Improving Neuroplasticity and Lower Limb Function after Chronic Stroke: A Pilot Study

**DOI:** 10.3390/biology12040515

**Published:** 2023-03-29

**Authors:** Hsien-Lin Cheng, Chueh-Ho Lin, Sung-Hui Tseng, Chih-Wei Peng, Chien-Hung Lai

**Affiliations:** 1Department of Physical Medicine and Rehabilitation, Taipei Medical University Hospital, Taipei 11031, Taiwan; 2International Ph.D. Program in Gerontology and Long-Term Care, College of Nursing, Taipei Medical University, Taipei 11031, Taiwan; 3Research Center in Nursing Clinical Practice, Wan Fang Hospital, Taipei Medical University, Taipei 110, Taiwan; 4Department of Physical Medicine and Rehabilitation, School of Medicine, College of Medicine, Taipei Medical University, Taipei 11031, Taiwan; 5School of Biomedical Engineering, College of Biomedical Engineering, Taipei Medical University, Taipei 11031, Taiwan; 6School of Gerontology and Long-Term Care, College of Nursing, Taipei Medical University, Taipei 11031, Taiwan

**Keywords:** stroke, repetitive transcranial magnetic stimulation, visual feedback training, motor-evoked potential, berg balance scale, timed up and go test

## Abstract

**Simple Summary:**

Sustained gait impairment is a common deficit and one of the causes of long-term disability after a stroke. Repetitive transcranial magnetic stimulation (rTMS) has shown promise for modulating cortical excitability over the leg region and enhancing activity plasticity in chronic stroke patients. This study shows amelioration of corticospinal excitability, balance, and functional mobility after a combination of rTMS and visual feedback training.

**Abstract:**

After a stroke, sustained gait impairment can restrict participation in the activities listed in the International Classification of Functioning, Disability, and Health model and cause poor quality of life. The present study investigated the effectiveness of repetitive transcranial magnetic stimulation (rTMS) and visual feedback training (VF) training in improving lower limb motor performance, gait, and corticospinal excitability in patients with chronic stroke. Thirty patients were randomized into three groups that received either rTMS or sham stimulation over the contralesional leg region accompanied by VF training groups in addition to the conventional rehabilitation group. All participants underwent intervention sessions three times per week for four weeks. Outcome measures included the motor-evoked potential (MEP) of the anterior tibialis muscle, Berg Balance Scale (BBS) scores, Timed Up and Go (TUG) test scores, and Fugl–Meyer Assessment of Lower Extremity scores. After the intervention, the rTMS and VF group had significantly improved in MEP latency (*p* = 0.011), TUG scores (*p* = 0.008), and BBS scores (*p* = 0.011). The sham rTMS and VF group had improved MEP latency (*p* = 0.027). The rTMS and VF training may enhance the cortical excitability and walking ability of individuals with chronic stroke. The potential benefits encourage a larger trial to determine the efficacy in stroke patients.

## 1. Introduction

It is well known that the corticospinal tract is considered a crossed pathway, consistent with the clinical findings that injury to the motor cortex of one hemisphere after stroke results in contralateral paresis [1,2]. After a stroke, patients often experience lower limb motor deficits that interrupt their balance and gait functions, which leads to a higher risk of falls [3]. Sustained gait impairment results in restricted participation in the activities listed in the International Classification of Functioning, Disability, and Health (ICF) model and a poor quality of life [4]. Occupational therapy and physical therapy are the standards for stroke rehabilitation and motor recovery. Several researchers have developed new restorative therapies with a focus on neural activity in brain network dynamics. Motor recovery after stroke has been reported to be considerably influenced by the neuroplasticity of the brain motor network [5,6,7,8]. This reorganization of the brain continues throughout the patient’s life; it is particularly prominent in the acute phase after a brain lesion but can persist for years following a stroke [7,8].

Numerous studies have reported that after a stroke on the medial cerebral artery, there is a decrease in corticospinal excitability of the affected motor cortex [9,10,11]; generally, enhanced contralesional excitability and increased interhemispheric inhibition occur. This state is associated with poorer function in the paretic limbs [12,13,14]. Studies have demonstrated that cortical activity shifting from the contralesional to the ipsilesional motor area is associated with improved outcomes [10,13,14,15]. These findings indicate that modulating cortical excitability to restore normal neural activity patterns is a potential strategy for stroke rehabilitation.

Repetitive transcranial magnetic stimulation (rTMS) is a noninvasive therapeutic tool that can be used to modulate cortical excitability either directly through the application of facilitatory stimulation (high-frequency) over the lesioned hemisphere or indirectly through inhibitory stimulation (low-frequency) to the contralesional hemisphere [9,11,15]. Inhibitory (1-Hz) rTMS applied over the contralesional hemisphere is safe and involves hotspots that are easier to locate and were reported to increase excitability within the ipsilesional hemisphere [16]. Previous studies have demonstrated that 1-Hz rTMS improved walking ability as well as motor function and led to a more symmetrical gait pattern in patients after stroke [17,18,19]. In addition, a recovery of motor deficits was associated with reduced interhemispheric asymmetry in leg motor excitability [20]. Although these results are preliminary, they support the application of rTMS over the leg region in patients with stroke.

Visual feedback (VF) training is a common approach to stroke rehabilitation. VF systems involve computer-based technology that provides feedback on performance. Patients with a stroke can apply this feedback by repeating activities in training with increasing intensity [21]. The repetition facilitates motor learning and neuroplasticity [22]. Studies have demonstrated that VF training in tandem with visual, auditory, and proprioceptive feedback can effectively improve muscle activation, balance, and walking ability [23,24] and is associated with an increase in activity in many regions within the visuomotor network and the ipsilesional primary motor cortex [25,26,27]. The ankle movement is crucial in the recovery of gait function after stroke [12,13]. In our previous study, we proposed a novel ankle joint motion- and position-sensing measurement system that can be used to measure the range of motion and proprioception of the ankle [28]. We incorporated this system into a video game-based training program in which patients with ankle inversion and eversion performed ankle dorsiflexion and plantarflexion in the sagittal and frontal planes [28]. In other studies, VF gaming training was conducted in 12 to 18 sessions with intervention periods of 4 to 6 weeks [22,29,30,31].

Although numerous studies have demonstrated that rTMS or VF training is effective in improving motor recovery after stroke, no study has combined rTMS and VF training of the lower limbs for stroke rehabilitation. The present study investigated the effects of combinations of rTMS and VF training, sham rTMS and VF training, and sham rTMS and conventional training on lower limb motor performance, gait function, and corticospinal excitability in patients with chronic stroke. We selected the tibialis anterior (TA) muscles as the targets for rTMS because they are crucial for ankle joint movement and are located in a focus region of our VF training system. Moreover, this study examined the efficacy of VF training on lower limb function, balance, mobility, and corticospinal excitability in individuals with chronic stroke and compared it to the effects of conventional rehabilitation. We hypothesize that the patients receiving real rTMS plus VF training would exhibit greater improvement in their lower limb motor function, balance, and mobility and reduction in contralesional to ipsilesional interhemispheric inhibition than those receiving sham rTMS plus VF training or sham rTMS plus conventional training would. In addition, we propose that the effectiveness of VF intervention on lower limb motor performance, gait function, and corticospinal excitability would be better than those undergoing conventional training in individuals with chronic stroke.

## 2. Materials and Methods

### 2.1. Participants

This study was approved by the Taipei Medical University Institutional Review Board (TMU-JIRB No.: N201607042). All participants provided written informed consent prior to participation. Patients with first-ever (absence of previous brain damage that caused motor problems in the lower limbs), chronic (>6 months after stroke onset), or monohemispheric stroke (e.g., infarction in medial cerebral artery territory) and who exhibited substantial leg impairment, as indicated by a Brunnstrom score above III, were enrolled in the study between 2017 and 2019. All participants were aged between 55 and 79 years and were able to walk independently for at least 10 m with or without assistive devices (e.g., cane or ankle-foot splint). The exclusion criteria were age over 80 years; a history of seizures or epilepsy; use of a pacemaker, aphasia, apraxia, concomitant neurological diseases, or other severe medical diseases; and undetectable motor-evoked potential (MEP) of the TA muscle of the nonparetic leg.

### 2.2. Study Design

The study design was blinded and blocked randomization. Patients were randomized into three matched groups: (1) Group E1, which received a 40-min of VF training immediately after a 10-min rTMS. (2) Group E2, which received a 40-min of VF training immediately after a 10-min sham rTMS. (3) Group C, which underwent a 40-min of conventional training immediately after a 10-min sham rTMS (Figure 1). All participants received the treatment course three times per week for 4 weeks. A well-trained and qualified occupational therapist delivered the rTMS and VF or conventional training. Measurements were taken in a pretest (1 day before intervention) and a posttest (1 day after intervention) by a blinded examiner. The protocols of the interventions applied to the three groups are presented in Figure 2.

### 2.3. Transcranial Magnetic Stimulation Procedure

Participants were seated comfortably, with a headrest to keep their heads stabilized and a leg rest to keep their knees flexed at 45° (Figure 3A). The MEP of the TA muscle was induced through single-pulse transcranial magnetic stimulation by using a MagStim Rapid2 stimulator (MagStim, Carmarthenshire, UK) with a 70-mm figure-eight-coil placed over the contralateral motor cortex and equipped with a surface electromyography (EMG) recording system (Sierra Wave EMG/EP system using Ag-AgCl electrodes). An active electrode was placed on the TA muscle, and a reference electrode was placed on the inferior border of the patella. The intensity was initially set at 100% of the machine output to determine the optimal stimulation site (hotspot). The optimal scalp position was determined by holding the coil tangentially over the leg area and slowly moving in 5-mm steps every 5–8 s along the optimal site for receiving a response from the TA muscle. Hotspots were defined as the sites that yielded the greatest TA MEP [18]. Next, we decreased the intensity in a stepwise manner while stimulating the hotspot. The motor threshold was defined as the minimal intensity required to evoke an MEP greater than 50 uV in more than 5 out of 10 trials during activation [32]. The location of the hotspot stimulation site of each hemisphere was marked and recorded to ensure consistency across sessions. The latency and amplitude of the MEP were measured in the pretests and posttests to identify changes in the corticospinal excitability. During each session, rTMS was performed using a 70-mm figure-eight coil at a 110% resting motor threshold and a train of 600 pulses (1 Hz) for 10 min over the leg area of the motor cortex on the unaffected hemisphere. Sham rTMS was performed with the coil held perpendicularly to the scalp and the same stimulus intensity and pattern.

### 2.4. Individualized Game-Based VF Intervention

To enable the provision of appropriate real-time VF training, we developed an individualized ankle haptic exercise program combined with a flying video game in which an airplane was controlled by the patient’s paretic ankle movements [28]. In the individualized ankle haptic exercise training group, each participant sat in a comfortable chair facing the training table, on which an LCD screen was placed. The ankle haptic interface was placed under the table. The paretic ankle was positioned on the ankle haptic interface, and Velcro was used to fix the paretic ankle’s position. The paretic leg was fixed using Velcro and lower extremity support to prevent abnormal compensation movements that might affect the paretic ankle VF training. The paretic joints were placed in a neutral starting and calibration position (Figure 3B). The apparatus was custom-made by Accu Balances Corporation (Taipei, Taiwan) and comprised an ankle haptic interface, two rotary potentiometers, and lower extremity support. Data on the movement of the paretic ankle joint in the sagittal and frontal planes were collected by two rotary potentiometers and were transferred to a computer as input device data from a Logitech USB compact stick (942-000009; Logitech International S.A., 1015 Lausanne, Switzerland) during training. Each participant was asked to move their paretic ankle to the extent of its range in the sagittal and frontal planes. The researcher then recalibrated the range of the joystick device input in the Windows operating system for each participant on the basis of these movement data before the training commenced. This ensured that the full range of the game controls would be adjusted to match each patient’s movement limitations. When the patients participated in the individualized game-based VF intervention, the movements of their paretic ankle controlled an aircraft in the flying video game, and this movement was displayed in real-time on the LCD monitor to provide the participants with direct VF.

### 2.5. Conventional Training

The conventional training involved lower extremity strengthening, transfer, balance, and functional ambulation training and was individualized to suit the functional status of each patient.

### 2.6. Outcome Measurements

The Berg Balance Scale (BBS) is used to objectively determine a patient’s ability to balance safely as they perform a series of tasks [33]. The BBS contains 14 items, each of which is rated on a 5-point ordinal scale ranging from 0 to 4, with 0 representing the lowest level of function and 4 representing the highest level of function. The Time Up and Go (TUG) test, which is used to measure dynamic balance and an individual’s ability to perform advanced mobility tasks [34], was used to identify changes in the gait and balance of the study participants. If the patient was unable to complete the TUG test within 120 s, their time was recorded as 120 s. Neurological recovery of the lower limbs was assessed using the Fugl–Meyer Assessment of Lower Extremity (FMA-LE) [35]. Each item was rated on a 3-point ordinal scale ranging from 0 (no performance) to 2 (complete performance), with the highest possible score of 34. The Fugl–Meyer Assessment is a feasible and efficient clinical examination that has been recommended for evaluating changes in motor impairment after stroke.

MEP has been demonstrated to be a sensitive measure for analyzing residual corticospinal functions and a predictor of motor recovery after stroke. In the present study, in the analysis of MEP, the optimal single TMS settings were adjusted to obtain the highest MEP. The stimulation intensity was set to 110% of the initial resting motor threshold. The highest MEP (hotspot) was determined, and the peak-to-peak MEP amplitudes and MEP latencies from 10 motor responses induced at an intensity of 110% of the initial resting motor threshold were averaged. If the TA muscles were unresponsive at the resting motor threshold, the MEP was recorded as undetectable.

### 2.7. Data Analysis

We used SPSS software (version 19.0) for data analyses. The χ^2^ analysis was used to compare the categorical demographic variables. Change scores were from the baseline calculated by subtracting pretest data from posttest data. All continuous variables (BBS scores, TUG scores, FMA-LE scores, MEP latency, MEP amplitude, and change) were tested using the Kruskal–Wallis test to identify intergroup differences in the baseline characteristics, pretest results, and posttest results. To investigate the effects of the intervention, we adopted the nonparametric Wilcoxon signed-rank test to identify intragroup differences in the pretest and posttest scores. The nonparametric Mann–Whitney U test was used to compare the cortical excitability of the bilateral hemispheres. Significance was set as a 2-tailed *p* < 0.05.

## 3. Results

### 3.1. Participants

This study enrolled 30 patients with first-ever monohemispheric or chronic stroke. No significant differences were identified in the baseline characteristics between the rTMS and VF, sham rTMS and VF, and sham rTMS and conventional rehabilitation groups (Table 1). The average time after the onset of stroke was 36.4 months, and the average Brunstrom stage of the paretic leg was 3.7, which indicated that the participants had chronic and profound lower limb motor deficits. None of the participants reported seizure induction, dizziness, or adverse events after the rTMS or sham stimulation.

### 3.2. Motor Performance

The motor performance results and functional measurements are listed in Table 2. No significant intergroup differences were identified in the pretest and the posttest, with the exception of Group E1, which had significantly different BBS (Z = −2.539, *p* = 0.011) and TUG (Z = −2.666, *p* = 0.008) scores in the posttest.

### 3.3. Corticospinal Excitability

The corticospinal excitability results are presented in Table 3. The latency and amplitude of the MEP of the TA muscle on the affected side were undetectable in 11 patients during the pretest (4 in Group E1, 4 in Group E2, and 3 in Group C). No significant intergroup differences were noted in the pretest. After the intervention, we discovered a non-significant intergroup difference in increased MEP latency in the TA muscle on the unaffected side (*p* = 0.092). However, it was significantly increased in Group E1 (Z = −2.547, *p* = 0.011) but not in Group E2 and Group C. In contrast, a significant intergroup difference in increased MEP amplitude in the TA muscle of the affected side (F = 4.438, *p* = 0.006), and it was significantly increased in Group E1 but not in Group E2 and Group C (X = −2.207, *p* = 0.027). These findings indicated that the 1-Hz rTMS over the unaffected hemisphere modulated the corticospinal excitability of both hemispheres. Notably, in one patient in Group E1 who was determined to be unresponsive in the pretest, both MEP latency and amplitude were detected on the affected side in the posttest.

In the pretest, the interhemispheric differences in MEP latency were significant for all three groups (Group E1: Z = −3.256, *p* = 0.001; Group E2: Z = −2.397, *p* = 0.017; Group C: Z = −2.832, *p* = 0.005; Figure 4A). The MEP amplitudes had an obvious tendency to show interhemispheric differences in all groups, particularly significant differences in Group C (Group E1: Z = −0.875, *p* = 0.382; Group E2: Z = −1.877, *p* = 0.060; Group C: Z = −2.664, *p* = 0.008; Figure 4B). After the intervention, the patients in Group E1 exhibited relatively symmetrical MEP latency between the bilateral hemispheres (Group E1: Z = 0.537, *p* = 0.591; Group E2: Z = −2.279, *p* = 0.023; Group C: Z = −3.065, *p* = 0.002; Figure 4C). No significant interhemispheric differences in MEP amplitude were noted for any of the three groups in the posttest, although Group E1 exhibited a trend of improvement in the affected hemisphere (Group E1: Z = −0.736, *p* = 0.462; Group E2: Z = −2.081, *p* = 0.051; Group C: Z = −2.984, *p* = 0.003; Figure 4D).

## 4. Discussion

The present study investigated the effectiveness of contralesional rTMS combined with subsequent VF intervention in patients with chronic stroke. To the best of our knowledge, this is the first study to investigate the effectiveness of combining rTMS and VF training of the lower limbs for individuals with chronic stroke; previous studies have investigated whether rTMS or VF training could improve upper limb motor function or motor activity after stroke [36,37]. Our results reveal that none of the observed outcome variables differed significantly among the three groups. However, the group that completed the rTMS and VF training treatment exhibited within-group significant differences in their BBS scores, TUG scores, and MEP latencies and amplitudes. The findings of the present study are consistent with the previous studies and indicate that rTMS can be used to induce neuroplastic changes and promote motor function restoration [19,20].

MEP, which can be generated through the application of single-pulse TMS to the motor cortex, can be used to quantify corticospinal excitability during stimulation [38]. In the current study, we discovered a significantly prolonged MEP latency and a trend of a decreasing MEP amplitude in the unaffected hemisphere in Group E1 (the rTMS and VF training group) that were not present in the other groups. Our results also reveal that Group E1 exhibited an obvious increase in MEP amplitude in the affected hemisphere after the intervention. These results indicate that the 1-Hz rTMS applied to the unaffected side of the brain inhibited corticospinal excitability in the unaffected hemisphere and enhanced corticospinal excitability in the affected hemisphere. The findings of this study support those of Wang et al. [20], whose data revealed that inhibitory (1-Hz) rTMS reduced MEP in the unaffected hemisphere and consequently increased the MEP amplitude in the affected hemisphere. Hence, our results are compatible with the concept of interhemispheric competition because they indicate that the 1-Hz rTMS administered to the unaffected hemisphere reduced the interhemispheric inhibition of the affected hemisphere [39,40]. Nevertheless, the inhibitory/excitatory effect of rTMS might depend on anatomical inter-individual variability of the corticospinal tract [41,42]. In particular, the ipsilateral uncrossed corticospinal tract is an implication for motor recovery [2,39,40]. In the present study, this mechanism would be interlaced with the VF training over affected muscles [2,41]. However, it is necessary to investigate further the exact mechanism of the combination of rTMS and VF training.

VF training is often considered to be an effective approach to rehabilitation because it offers repetitive, intensive, and meaningful task-specific training that promotes cortical reorganization. [43,44,45]. Several studies have reported that ankle and foot abnormalities can have subsequent effects on the knee and hip joints as well as on gait patterns [46,47,48]. In the present study, the participants exhibited significant improvements in their BBS and TUG scores after they completed the rTMS and VF training. Although Group E2, which received sham rTMS and VF training, had improvement in BBS and TUG scores after the intervention, the intragroup differences and intergroup differences did not reach significance. This finding would be due to the small sample of patients recruited and the duration and intensity of training for this study.

The lower limb motor function was also evaluated by the FMA-LE test. According to the FMA-LE scores, the patients enrolled in this study featured motor disabled impairment that was classified as moderate (scores between 20 and 28) related to chronicity and/or severity of stroke [49]. The participants of this study have been onset for more than 26 months after stroke. Further study is required to determine the duration and intensity of intervention. Moreover, both the VF training and conventional training groups exhibited improvement in their FMA-LE, BBS, and TUG scores and their corticospinal excitability after the interventions. However, these improvements were modest because no difference among the groups was found. These findings are compatible with those of other studies, which have reported results that demonstrated that interactive video game exercises had similar effects to conventional rehabilitation on upper limb function, gait speed, balance, participation, and quality of life [21,29,50].

This pilot study has several limitations, including a small sample size and short intervention duration, which precluded long-term evaluation and without the follow-up screen. In addition, a figure-eight coil was used to apply the stimulator to the site. Nondetectable MEP was recorded for 11 of the 30 included patients; a cone coil may be able to induce MEP more easily and achieve more effective therapy. Nevertheless, the rTMS treatment of this study was demonstrated to positively affect the participants’ BBS scores, TUG scores, and corticospinal excitability. Other studies using the same parameters and coils as those used in this study have also reported that post-rTMS treatment improved the short-term outcomes of stroke survivors, including leg impairment, mobility, and corticospinal excitability [18,20].

## 5. Conclusions

The present study revealed that combined rTMS and VF training of the paretic ankle might modulate corticospinal excitability and subsequently potentially improve balance and functional mobility in individuals with chronic stroke. However, the pilot results of this study should be further validated in future studies with larger sample sizes or longer clinical trials. In addition, investigation of the mechanism of the combination of rTMS and VF therapy in stroke recovery should take into account the possibility of involvement of ipsilateral corticospinal projection and inter-hemispheric commissural connections. This exact mechanism is necessary for further study.

## Figures and Tables

**Figure 1 biology-12-00515-f001:**
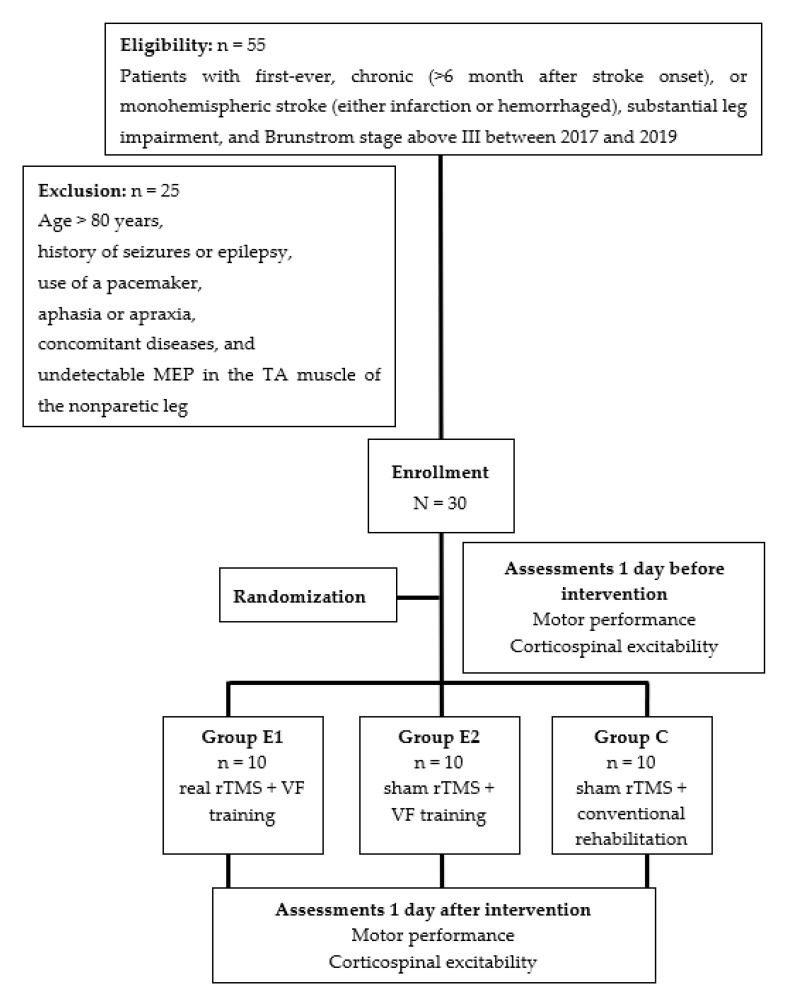
Enrollment flowchart.

**Figure 2 biology-12-00515-f002:**
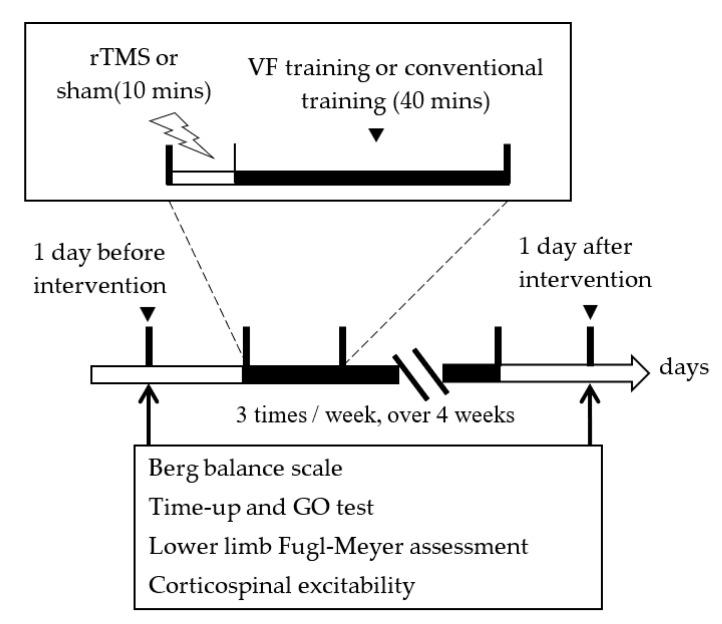
Study design.

**Figure 3 biology-12-00515-f003:**
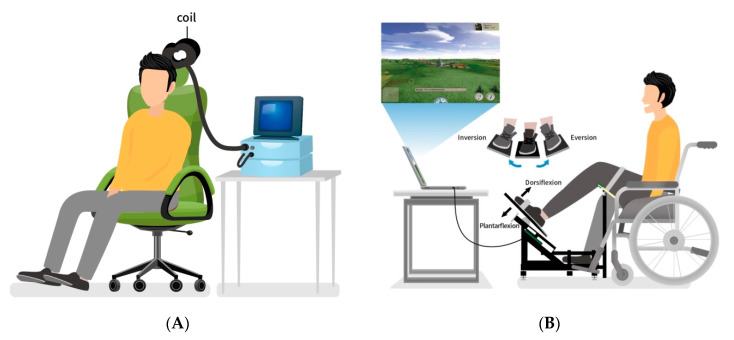
The rTMS and the VF training. (**A**) Application of 1-Hz, 110% resting motor threshold rTMS or sham rTMS before visual feedback training or conventional rehabilitation. (**B**) Architecture of game-based visual feedback intervention system; movement at the extent of the paretic ankle’s range in the sagittal (plantarflexion/dorsiflexion) and frontal (eversion/inversion) planes used to move an aircraft displayed in real-time on an LCD monitor to provide direct visual feedback.

**Figure 4 biology-12-00515-f004:**
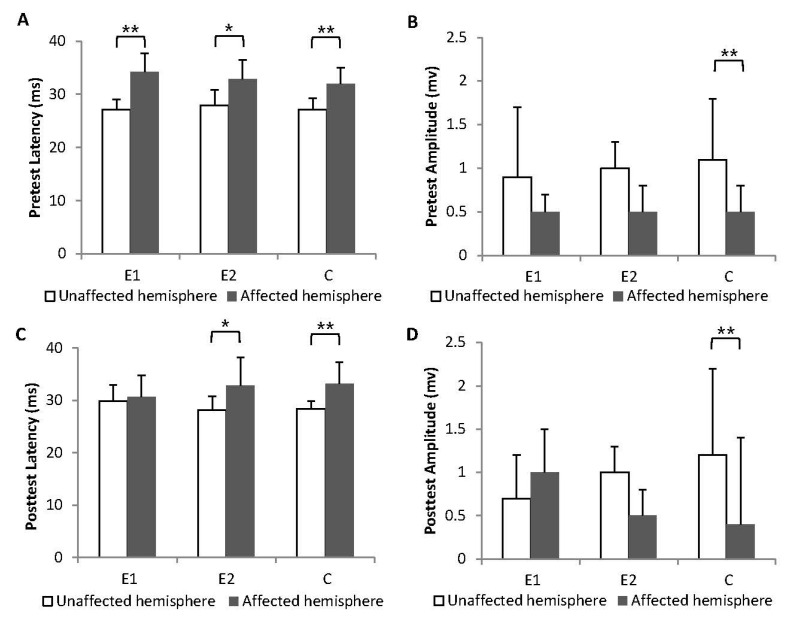
MEP latencies and amplitudes of unaffected and affected hemispheres. (**A**) MEP latencies of unaffected and affected hemispheres in the pretest. (**B**) MEP amplitudes of unaffected and affected hemispheres in the pretest. (**C**) MEP latencies of unaffected and affected hemispheres in the posttest. (**D**) MEP amplitudes of unaffected and affected hemispheres in the posttest. Error bars represent the standard deviation of the mean. * *p* < 0.05, ** *p* < 0.01.

**Table 1 biology-12-00515-t001:** Participant characteristics.

	Group E1 (*n* = 10)	Group E2 (*n* = 10)	Group C (*n* = 10)	F (*r*)	** *p* **
Age, year	62.3 ± 15.3	56.4 ± 17.5	61.1 ± 13.2	0.406	0.670
Sex, male/female	6/4	6/4	7/3	(0.287)	0.866
Hemispheric side, left/right	6/4	5/5	4/6	(0.800)	0.670
Modified Ashworth Scale, MAS	0.8 ± 0.9	0.9 ± 0.8	1.2 ± 1.0	0.485	0.621
Mini-Mental State Examination	29.9 ± 0.3	30 ± 0.1	29.6 ± 0.9	1.258	0.300 *
Time poststroke, months	29.8 ± 20.9	31.6 ± 23.8	48.0 ± 29.0	1.635	0.214 *
Br. Stage ^1^ of lower extremity	3.8 ± 0.8	3.7 ± 0.5	3.5 ± 0.8	0.444	0.646

^1^ Br. stage, Brunnstrom stage. Values are expressed as means ± standard deviations. Intergroup differences were analyzed using Kruskal–Wallis test for continuous variables and the χ^2^ test for categorical variables. * *p* < 0.05.

**Table 2 biology-12-00515-t002:** Motor performance and functional measurements.

	Pretest	Posttest	Change	*P*^a^ for Intragroup Difference	*P*^b^ for Intergroup Difference
Fugl–Meyer Assessment of Lower Extremity (FMA-LE)					
Group E1	25.1 ± 9.2	25.6 ± 8.9	0.5 ± 0.8	0.102	0.441
Group E2	19.7 ± 8.7	20.6 ± 8.9	0.9 ± 2.1	0.102	
Group C	24.9 ± 7.1	25.0 ± 7.0	0.1 ± 0.3	0.317	
Berg Balance Scale (BBS)					
Group E1	41.7 ± 11.5	43.6 ± 10.9	1.9 ± 1.5	0.011 *	0.167
Group E2	34.7 ± 13.8	37.5 ± 14.9	2.8 ± 4.6	0.066	
Group C	40.3 ± 18.2	41.4 ± 18.5	1.1 ± 2.5	0.109	
Time Up and Go (TUG)					
Group E1	39.3 ± 32.2	29.8 ± 17.2	−9.4 ± 17.8	0.008 *	0.052
Group E2	51.4 ± 40.2	49.1 ± 40.2	−2.2 ± 4.3	0.093	
Group C	30.8 ± 23.9	30.5 ± 39.4	−0.2 ± 1.6	0.541	

Values are expressed as means ± standard deviations. *P*^a^ Wilcoxon signed-rank test. *P*^b^ Kruskal–Wallis test. * *p* < 0.05.

**Table 3 biology-12-00515-t003:** Cortical excitability.

	Pretest	Posttest	Change	*P*^a^ for Intragroup Difference	*P*^b^ for Intergroup Difference
MEP latency ^UH^, ms					
Group E1	27.1 ± 1.9, n = 10	29.8 ± 3.1, n = 10	2.6 ± 3.5	0.011 * (−2.547)	0.092
Group E2	27.9 ± 2.9, n = 10	28.1 ± 2.7, n = 10	0.2 ± 3.0	0.799 (−0.255)	
Group C	27.1 ± 2.2, n = 10	28.4 ± 1.5, n = 10	1.2 ± 1.9	0.093 (−1.682)	
MEP amplitude ^UH^, mV					
Group E1	0.9 ± 0.8, n = 10	0.7 ± 0.5, n = 10	−0.1 ± 0.3	0.205 (−1.268)	0.546
Group E2	1.0 ± 0.3, n = 10	1.0 ± 0.3, n = 10	0.02 ± 0.5	0.758 (−0.308)	
Group C	1.1 ± 0.7, n = 10	1.3 ± 1.0, n = 10	0.1 ± 0.7	0.989 (−0.001)	
MEP latency ^AH^, ms					
Group E1	34.2 ± 3.5, n = 6	30.7 ± 4.1, n = 7	0.9 ± 12.8	0.116 (−1.572)	0.413
Group E2	32.9 ± 3.6, n = 6	32.8 ± 5.4, n = 6	−0.03 ± 4.8	0.917 (−0.105)	
Group C	32.0 ± 3.0, n = 7	33.2 ± 4.1, n = 7	0.8 ± 3.9	0.674 (−0.420)	
MEP amplitude ^AH^, mV					
Group E1	0.5 ± 0.2, n = 6	1.0 ± 0.7, n = 7	0.3 ± 0.5	0.027 * (−2.207)	0.006 *
Group E2	0.5 ± 0.2, n = 6	0.5 ± 0.5, n = 6	−0.03 ± 0.2	0.581 (−0.552)	
Group C	0.5 ± 0.3, n = 7	0.4 ± 0.3, n = 7	−0.06 ± 0.2	0.343 (−0.948)	

Values are expressed as means ± standard deviations. MEP, moto-evoked potential; UH, unaffected hemisphere; AH, affected hemisphere. *P*^a^ Wilcoxon signed-rank test. *P*^b^ Kruskal–Wallis test. * *p* < 0.05.

## Data Availability

Not applicable.

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
