# Peer review of "Effectiveness of Repetitive Transcranial Magnetic Stimulation Combined with Visual Feedback Training in Improving Neuroplasticity and Lower Limb Function after Chronic Stroke: A Pilot Study"

_biology, 2023, doi:10.3390/biology12040515_

Round 1

Reviewer 1 Report

In this experiment, 1-Hz (low-frequency) rTMS was performed over the primary motor cortex (M1) of the unaffected hemisphere of N=30 stroke patients. MEP recording was performed over the ankle muscle of both legs for test-retest comparison before and after a new combined protocol of visual-feedback training of the paretic ankle and neuro stimulation of the unaffected hemisphere. The results show that 1-Hz rTMS protocol over the unaffected hemisphere modulated CSE of both hemispheres.

I liked very much reading the paper and evidence on the neural plasticity after the combined treatment are very clear. It will offer an interesting contribute to the study of stroke neurorehabilitation. I have some suggestions that may further improve its conclusiveness.

p. 2, line 42. As a first step, it is necessary to give a definition of stroke and specify in which cases a stroke affects the primary motor cortex, thus creating dysfunction in the limbs contralateral to the injured hemisphere - where the crossed corticospinal tract carries information to the contralateral interneurons and motor neurons.

p. 2, line 54. Perhaps it would be clearer to say that after a stroke on the medial cerebral artery there is a decrease in corticospinal excitability of the affected motor cortex.

p. 2, Line 66. When you talk about 1-Hz rTMS you have to explain its specific effect: 1-Hz rTMS is considered to have an inhibitory effect in healthy people because it suppresses the excitability of the motor cortex. However, the effect of 1 Hz rTMS might not be inhibitory for everyone and the concept of inhibitory/excitatory effect of rTMS has been revised (Caparelli et al., 2012). Moreover, recent research has shown that also anatomical inter-individual variability of the corticospinal tract can modulate the corticospinal excitability (Betti et al., 2022). These findings must be acknowledged before drawing any conclusions.

Caparelli E, Backus W, Telang F, Wang G, Maloney T, Goldstein R, Henn F. Is 1 Hz rTMS Always Inhibitory in Healthy Individuals? Open Neuroimag J. 2012;6:69-74. Epub 2012 Jul 17. PMID: 22930669; PMCID: PMC3428632.

Betti, S., Fedele, M., Castiello, U. et al. Corticospinal excitability and conductivity are related to the anatomy of the corticospinal tract. Brain Struct Funct 227, 1155–1164 (2022).

p. 2, Line 82. “and the ipsilateral primary motor cortex”. Ipsilateral to what?

p.3, line 96. If this is your hypothesis, and you do not expect any difference between those who receive rTMS and those who receive control sham, then it is unclear what purpose should be served by using rTMS. The same doubt arises from reading the results in the Abstract. I suggest rewriting the experimental hypotheses and results reported in the Abstract.

p.3, line 105. The criteria for inclusion of participants in this study should be better described. For example, one should make sure that only patients with focal Medial Cerebral Artery injury and absence of previous brain damage that caused motor problems in the lower limbs should be included in the study.

p.3, line 115. Where the participants matched for hemispheric side of the lesion? This info is not reported in Table 1.

p.3, line 124. In order to affirm the validity of a new rehabilitation protocol, generally the practice involves four evaluation sessions:

 S1. Baseline Pre-Treatment

 S2. Post-Treatment Evaluation

 S3. Short-term follow-up (e.g., 3 months from Baseline Pre-Treatment)

 S4. Long-term follow-up (e.g., 6 months from Baseline Pre-Treatment)

 Have you planned any form of follow-up?

p.3, line 126. “The MEP was induced through TMS”. Please, be more specific. MEPs are evoked by single-pulse TMS (spTMS), which is different from rTMS. Both techniques should be clearly described.

p. 3, line 134. What coordinates were used to locate the ankle muscle hot spot?

p.3, line 143. What is the intensity of stimulation adopted in the protocol? Here it reads 100% of the resting motor threshold. But elsewhere it reads 110% (line 205).

P4., line 214. Please define the dependent variable “change”.

Figure 4. In order to compare MEP amplitude data between pre- and post-test, the Y-axes of graphs B and D must show the same measurement scale.

p.6, line 197. The neurologic recovery of the lower limbs was also assessed using the FMA-LE scale. What is the interpretation for the null result in the post-test output of this scale? And for the interhemispheric difference in terms of MEP amplitude at the pre-test phase only in the C group?

p. 8, line 244. The intergroup difference should be better described: it refers to MEP amplitude when spTMS was performed over the affected hemisphere. This is a crucial result because the 1-Hz rTMS protocol over the unaffected hemisphere modulated CSE of both hemispheres.

Conclusions

Notably, neuroimaging research has suggested a role of the contralesional hemisphere in promoting recovery after stroke through the ipsilateral uncrossed corticospinal tract (CST) fibers descending to ipsilateral spinal segments. In the present study, this mechanism would be interlaced with the VF training over the same (affected) muscles. This aspect should be stated, without detracting from the validity of your paradigm.

Alawieh, A., Tomlinson, S., Adkins, D., Kautz, S., & Feng, W. (2017). Preclinical and clinical evidence on ipsilateral corticospinal projections: implication for motor recovery. Translational stroke research, 8, 529-540.

Jankowska, E., & Edgley, S. A. (2006). How can corticospinal tract neurons contribute to ipsilateral movements? A question with implications for recovery of motor functions. The Neuroscientist, 12(1), 67

Round 2

Reviewer 1 Report

The Authors have addressed all my concerns satisfactorily, then I feel the manuscript is ready to be published